# ON THE CONVERGENCE DIRECTION OF GRADIENT DESCENT

**Shuo Chen**
Institute of Information Science
Beijing Jiaotong University
Beijing, China
schen1307@foxmail.com

**Xiaolong Li**
Institute of Information Science
Beijing Jiaotong University
Beijing, China
lixl@bjtu.edu.cn

**Jiaying Peng**
School of Mathematical Science
Capital Normal University
Beijing, China
jiayingpeng@cnu.edu.cn

**Yao Zhao**
Institute of Information Science
Beijing Jiaotong University
Beijing, China
yzhao@bjtu.edu.cn

## ABSTRACT

Gradient descent (GD) is a fundamental optimization method in deep learning, yet its asymptotic directional properties remain less understood. In this paper, we prove that if GD converges, its trajectory either aligns toward a fixed direction or oscillates along a specific line. The fixed-direction convergence occurs under small learning rates, while the oscillatory convergence behavior emerges for large learning rates. This result offers a new lens for understanding long-term GD dynamics. Experimentally, we find that this directional convergence behavior also appears in stochastic gradient descent and Adam. Furthermore, we discuss how these theoretical findings regarding oscillatory convergence might offer a perspective on the sharpness dynamics observed in the Edge of Stability (EoS) regime. Our work provides both theoretical clarity and practical insight into the behavior of dynamics for multiple optimization methods as well as EoS.

## 1 INTRODUCTION

Gradient descent (GD) is one of the most extensively studied optimization algorithms. While classical analysis typically focuses on small learning rates, the precise structure of the GD trajectory remains less understood. Recent empirical lines of research, such as the observations regarding the *Edge of Stability* (EoS) (Cohen et al., 2021), highlight that GD can exhibit complex, non-monotonic dynamics where the loss fluctuates over short timescales. Motivated by these rich dynamical behaviors, this work focuses on the asymptotic directional properties of GD, investigating theoretically how the trajectory aligns or oscillates relative to the learning rate.

A common framework for analyzing GD assumes that the function to be minimized is convex and $L$-smooth (Nesterov, 2003). Formally, let $f : D \subset \mathbb{R}^n \to \mathbb{R}$ be a differentiable function defined on a domain $D$, satisfying the $L$-smooth condition, i.e., for all $\boldsymbol{x}, \boldsymbol{y} \in D$,

$$\|\nabla f(\boldsymbol{x}) - \nabla f(\boldsymbol{y})\| \leq L\|\boldsymbol{x} - \boldsymbol{y}\|. \tag{1}$$

Conventionally, GD dynamics $\{\boldsymbol{x}_k\}_{k \geq 0}$ is defined as,

$$\boldsymbol{x}_{k+1} = F(\boldsymbol{x}_k), \tag{2}$$

where

$$F(\boldsymbol{x}) = \boldsymbol{x} - \eta \nabla f(\boldsymbol{x}). \tag{3}$$

By the classical descent lemma (Ahn et al., 2022), this framework ensures stable convergence and allows for a well-established convergence rate, if the learning rate satisfies $0 < \eta < 2/L$. However, the condition on $L$-smooth equation 1 requires that the constant $L$ is sufficiently large. As a result, this condition usually fails on neural networks.

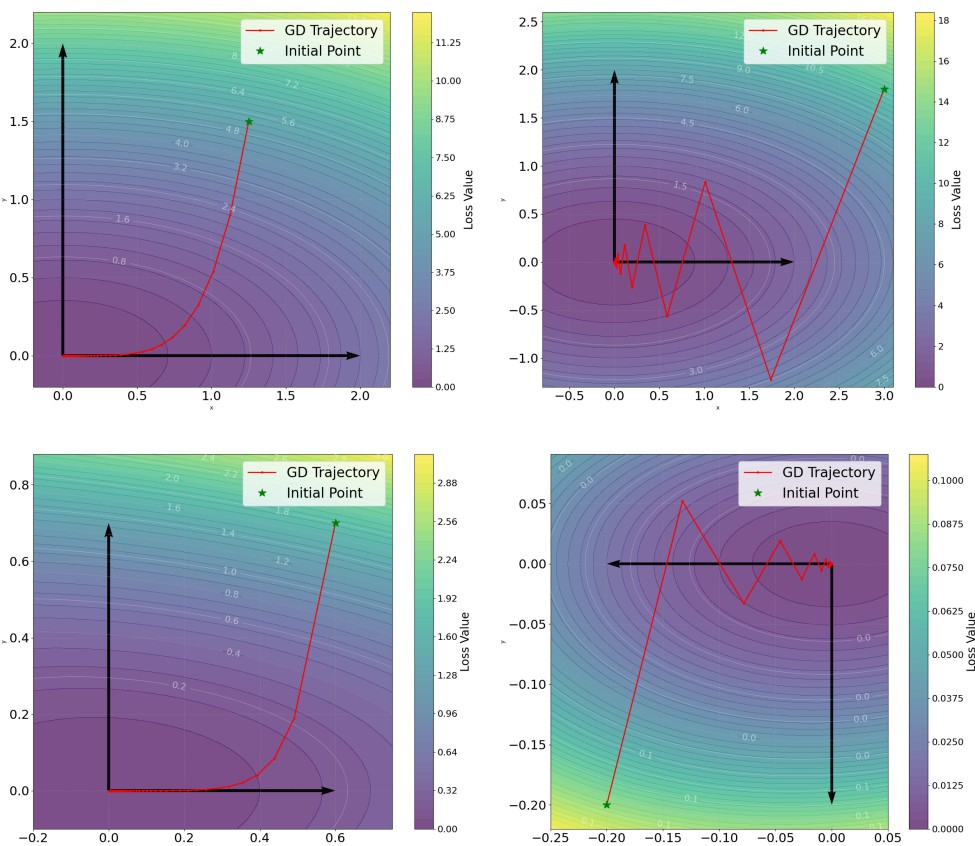

Figure 1: GD trajectories for $f(x, y) = x^2/2 + 2y^2$ with $\eta = 0.1$ (upper left, the convergence direction exists as $x$-axis) and $\eta = 0.42$ (upper right, the convergence direction alternates along $y$-axis), and for $f(x, y) = x^2/2 + 2y^2 + xy^2 + y^3$ with $\eta = 0.1$ (lower left, the convergence direction exists as $x$-axis) and $\eta = 0.42$ (lower right, the convergence direction alternates along $y$-axis).

Moreover, GD usually requires a learning rate satisfying $0 < \eta < 2/\lambda_n$ for convergence (Ahn et al., 2022), where $\lambda_n$ is the largest eigenvalue of the Hessian matrix at the local minimum. On the other hand, empirical results show that GD often continues to make progress even when the learning rate exceeds this bound at the first several GD iterations. More specifically, $2/\eta$ is smaller than the sharpness of $\boldsymbol{x}_k$ for the first several $\boldsymbol{x}_k$'s in the GD trajectory. Here, the sharpness is defined as the largest eigenvalue of the Hessian matrix at $\boldsymbol{x}_k$. This observation raises fundamental questions about the underlying mechanisms that govern GD's long-term behavior and convergence. Many works have been devoted to analyzing the GD dynamics from the perspective of stability, loss landscape, and optimization trajectories (Ahn et al., 2022; Zhu et al., 2023; Damian et al., 2023; Lee & Jang, 2023; Chen et al., 2024; Cai et al., 2024).

In this work, the convergence direction of GD is theoretically investigated. Specifically, depending on the learning rate, we prove that, if convergent, the GD trajectory either aligns with a fixed direction or oscillates along a specific line. To illustrate our idea, consider a simple convex quadratic function $f(x, y) = x^2/2 + 2y^2$, in which $(0, 0)$ is the unique global minimum. In this case, the GD dynamics are given by $x_k = (1 - \eta)^k x_0$ and $y_k = (1 - 4\eta)^k y_0$. Consider then the convergence direction, i.e., the limit of normalized vector

$$\boldsymbol{v}_k \triangleq \frac{(x_k, y_k)}{\|(x_k, y_k)\|}. \tag{4}$$

We have, for $x_0 \neq 0$ and $y_0 \neq 0$, $\lim_{k \to \infty} \boldsymbol{v}_k = (\text{sign}(x_0), 0)$ if $0 < \eta < 2/5$, and $\lim_{k \to \infty} (-1)^k \boldsymbol{v}_k = (0, \text{sign}(y_0))$ if $2/5 < \eta < 1/2$ (see Figure 1 for illustration). Moreover, such phenomenon also holds if small disturbance exists. For example, consider a quadratic function

with high-order disturbation, $f(x,y) = x^2/2 + 2y^2 + xy^2 + y^3$, The similar results arise up under the similar settings (see Figure 1 as well).

This result suggests that GD iterates align with a particular direction when approaching the minimum, rather than converging in arbitrary directions. Our analysis generalizes this observation by characterizing the exact convergence direction of the GD trajectory, providing a refined understanding of GD's asymptotic behavior.

Our key contributions are summarized as follows:

- **Convergence direction of GD:** We show that when GD converges, its trajectory admits a convergence direction that depends on the learning rate $\eta$. Specifically, if $0 < \eta < 2/(\lambda_1 + \lambda_n)$, the GD trajectory aligns toward a fixed direction. In contrast, if $2/(\lambda_1 + \lambda_n) < \eta < 2/\lambda_n$, the GD trajectory exhibits directional oscillations along a specific line. Here, $\lambda_1$ and $\lambda_n$ are the smallest and largest eigenvalues of the Hessian matrix at the local minimum respectively.

- **Insights for modern optimization methods:** We observe that popular optimization algorithms such as stochastic gradient descent (SGD) and Adam also exhibit similar alignment behaviors in practice. Our findings may inform the design of new optimization algorithms that explicitly exploit this directional alignment.

## 2 RELATED WORKS

A central theoretical foundation of our work builds on the celebrated proof of the *Gradient Conjecture*, originally posed by René Thom and proven in Parusinski et al. (2000). The conjecture concerns the behavior of gradient flow trajectories near critical points of real analytic functions. Specifically, consider a trajectory $\boldsymbol{x}(t)$ of the gradient vector field of a real analytic function $f$, with $\boldsymbol{x}(t) \to \boldsymbol{x}_0$ as $t \to \infty$. The Gradient Conjecture asserts that the normalized secants, $\lim_{t\to\infty} \frac{\boldsymbol{x}(t)-\boldsymbol{x}_0}{\|\boldsymbol{x}(t)-\boldsymbol{x}_0\|}$, converge. Actually, Parusinski et al. (2000) not only prove this conjecture but establishes a stronger statement: the radial projection of the trajectory onto the unit sphere has finite length. Their argument leverages Łojasiewicz inequalities, asymptotic critical values, and the construction of control functions. Notably, functions of the form $g(\boldsymbol{x}) = f(\boldsymbol{x})/r^l - a - r^\alpha$ grow fast enough along trajectories to guarantee directional convergence, where $r = \|\boldsymbol{x}\|$, $\alpha$ is a small positive constant, $l$ is a rational number, and $a$ is a negative constant. In Xing et al. (2018), the authors analyze the trajectory of SGD in neural networks and observe that consecutive gradient steps tend to follow a valley-like structure in the loss landscape, often moving in directions that alternate or oscillate. Their results highlight that the directionality of gradient steps carries meaningful geometric information about the optimization path, even across non-convex and noisy settings. In Morchdi et al. (2023), the authors investigate the phenomenon of gradient oscillation in neural network training by analyzing the correlation between consecutive gradient directions. They observe that a significant portion of optimization progress occurs during periods when consecutive gradient steps exhibit strong negative correlation, highlighting the complex and often non-monotonic behavior of gradient directions in practical training dynamics. Inspired by these works, we try to extend the continuous-time Gradient Conjecture to GD. We show that if convergent, the GD iterates exhibit analogous directional behavior, which either converge to a fixed direction or oscillate along a line. This connection provides a bridge between the geometry of continuous gradient flow and discrete optimization dynamics.

The phenomenon of EoS is an interesting topic in understanding GD dynamics. In Cohen et al. (2021), the authors first observe that GD frequently operates in a regime where the largest eigenvalue of the Hessian hovers around the critical threshold $2/\eta$, challenging traditional stability analyses. This discovery motivates further research into how optimization methods interact with sharpness and stability. A key extension of EoS appears in Sharpness-Aware Minimization (SAM) (Long & Bartlett, 2024). In that work, the authors show that SAM dynamically adjusts the Hessian norm via the gradient and perturbation radius, stabilizing training and promoting flatter minima. Moreover, Dai et al. (2024) discusses the impact of the batch normalization layer toward EoS. Recent work by Grimmer (2024) highlights that GD with periodically long steps can achieve provably faster convergence, despite violating descent at individual iterations.

Other works explore GD or SGD dynamics in multiple regimes under large learning rates (Hoffer et al., 2017; Li et al., 2019; Wu et al., 2023). Related studies examine curvature-aware learning rates

(Thomas et al., 2020; Wang & Ma, 2024; Cohen et al., 2024), and nonmonotone line search method (Fox et al., 2024), which adaptively adjust learning rates near the EoS without manual tuning. The works (Ma et al., 2022; Lee & Jang, 2023) further show that loss landscapes are subquadratic near minima and propose interaction-aware sharpness measures for mini-batch settings. At the same time, using momentum may modify GD's EoS behavior. In Phunyaphibarn et al. (2024), the authors show that Polyak's momentum leads to sharp curvature drops ("catapults") and raises the maximum stable sharpness, stabilizing training while promoting flatter minima. Another important direction is parameter averaging, and Nitanda et al. (2024) proves that averaged SGD smooths noisy updates and implicitly biases toward wider minima. More research on implicit bias can be found in Arora et al. (2022); Nacson et al. (2023). This complements EoS findings, as averaging helps mitigate the instability caused by large learning rates and sharp directions. Recent work also links EoS with multi-phase dynamics. For instance, Wang et al. (2022); Cai et al. (2024) observe that large learning rates in neural networks may induce the EoS phase and gradually more stable phases. In Yang et al. (2024), the authors show that untuned SGD can achieve optimal rates but is sensitive to unknown smoothness constants, leading to gradient explosions. In Wang & Ma (2024), the authors identify dynamic transitions in gradient flow training, suggesting EoS interacts with deeper phase structures in training. More studies on dynamic of different phases can be found in Damian et al. (2023).

Finally, the role of noise in SGD ties to EoS. For instance, Mulayoff et al. (2021) links large learning rates with smoother learned functions and shows that stable solutions depend on depth, supporting the regularization view of EoS. More research on the connection of geometry and dynamics can be found in Lee et al. (2016); Zhu et al. (2019); Martens (2020); Wang & Wu (2023). Overall, these studies show that normalization, momentum, averaging, and noise all shape the dynamics of training near EoS, highlighting the role of instability-driven behavior in modern optimization.

## 3 MAIN RESULTS

Consider a loss function $f \in C^3(\mathbb{R}^n)$ and its GD dynamics defined in equation 2 and equation 3 with learning rate $\eta$. For an isolated local minimum $\boldsymbol{x}^*$ of $f$, consider the GD trajectory $\{\boldsymbol{x}_k\}_{k \geq 0}$ converging to $\boldsymbol{x}^*$. Specifically, we define $V_\eta = \{\boldsymbol{x}_0 \in \mathbb{R}^n : \lim_{k \to \infty} \boldsymbol{x}_k = \boldsymbol{x}^*\}$. Moreover, assume that for any subset $W \subset \mathbb{R}^n$ with zero measure, $F^{-1}(W)$ also has zero measure, where $F$ is defined in equation 2. The zero measure condition of GD dynamics is a conventional assumption for technical reasons (Ahn et al., 2022; Chen et al., 2024). Then, our main result can be summarized as the following theorem.

**Theorem 1:** For the above defined function $f$, suppose the eigenvalues $\{\lambda_1, \ldots, \lambda_n\}$ of $\nabla^2 f(\boldsymbol{x}^*)$ satisfy $0 < \lambda_1 < \lambda_2 \leq \cdots \leq \lambda_{n-1} < \lambda_n$, for almost all initial points $\boldsymbol{x}_0 \in V_\eta$,

- If $0 < \eta < 2/(\lambda_1 + \lambda_n)$, the convergence direction of GD exists, i.e., $\lim_{k \to \infty} \frac{\boldsymbol{x}_k - \boldsymbol{x}^*}{\|\boldsymbol{x}_k - \boldsymbol{x}^*\|}$ exists.

- If $2/(\lambda_1 + \lambda_n) < \eta < 2/\lambda_n$, the alternative convergence direction of GD exists, i.e., $\lim_{k \to \infty} (-1)^k \frac{\boldsymbol{x}_k - \boldsymbol{x}^*}{\|\boldsymbol{x}_k - \boldsymbol{x}^*\|}$ exists.

In this theorem, the condition for eigenvalues means that $f$ is locally a strong convex function at $\boldsymbol{x}^*$. Moreover, for given $\eta > 0$, the initial point set $V_\eta$ is an open subset of $\mathbb{R}^n$ (Chen et al., 2024). Next, we proceed to sketch the proof and the complete proof is provided in the appendics.

### 3.1 PROOF SKETCH WITH $0 < \eta < 2/(\lambda_1 + \lambda_n)$

Without loss of generality, we assume $\boldsymbol{x}^* = \boldsymbol{0}$. Furthermore, we may assume that the Hessian matrix $\nabla^2 f(\boldsymbol{0})$ is diagonal (see Appendix A for details), i.e., $\nabla^2 f(\boldsymbol{0}) = \text{diag}(\lambda_1, \ldots, \lambda_n)$. Define

$$a = 1 - \eta\lambda_1, \qquad b = \max_{2 \leq i \leq n} \{|1 - \eta\lambda_i|\}. \tag{5}$$

Since $\eta < 2/(\lambda_1 + \lambda_n) < 2/2\lambda_1 = 1/\lambda_1$, it follows that $a > b \geq 0$. Let $x_{k,i}$ be the $i$-th component of $\boldsymbol{x}_k$, then the GD iteration can be expressed componentwisely as, for each $1 \leq i \leq n$,

$$x_{k+1,i} = x_{k,i} - \eta\partial_i f(\boldsymbol{x}_k) = (1 - \eta\lambda_i)x_{k,i} + g_i(\boldsymbol{x}_k), \tag{6}$$

where $g_i$ is defined by, for $\boldsymbol{x} = (x_1, ..., x_n) \in \mathbb{R}^n$,

$$g_i(\boldsymbol{x}) = \eta(\lambda_i x_i - \partial_i f(\boldsymbol{x})). \tag{7}$$

For simplicity, denote $g_{k,i} = g_i(\boldsymbol{x}_k)$ throughout the following proof. Clearly, $g_i \in \mathcal{C}^2(\mathbb{R}^n)$ and thus $g_{k,i}$ is a twicely differentiable function with respect to $\boldsymbol{x}_k$. Moreover, by GD iteration equation 2, $g_{k,i}$ can be regarded as a twicely differentiable function of $\boldsymbol{x}_0$. By the above preparation, we first establish a forward-invariant set for GD and provide key estimates for $g_{k,i}$. These results are summarized in the lemma below.

**Lemma 1**: For the function $f$ defined in Theorem 1, there exists an open set $\Omega \subset V_\eta$ containing $\boldsymbol{x}^* = \boldsymbol{0}$ and a constant $C > 0$ such that for any initial point $\boldsymbol{x}_0 \in \Omega$, the following properties hold,

- For all $k \geq 0$, $\boldsymbol{x}_k \in \Omega$.
- For all $k \geq 0$ and $1 \leq i \leq n$, $|g_{k,i}| \leq C\|\boldsymbol{x}_k\|^2$.
- For all $k \geq 0$ and $1 \leq i, j \leq n$, $\left|\frac{\partial g_{k,i}}{\partial x_{k,j}}\right| \leq C\|\boldsymbol{x}_k\|$.

The proof of Lemma 1 draws upon the main theorem derived in Chen et al. (2024), which is stated below for clarity.

**Theorem 2**: [Theorem 2, Chen et al. (2024)] Let $f \in \mathcal{C}^2(\mathbb{R}^n)$ and $\boldsymbol{x}^*$ is a local minimum. Suppose that there exists $r > 0$ such that $f(\boldsymbol{x}) > f(\boldsymbol{x}^*)$ holds for any $\boldsymbol{x}$ satisfying $\|\boldsymbol{x} - \boldsymbol{x}^*\| = r$. Moreover, assume that $\nabla f(\boldsymbol{x}) \neq 0$ for all $\boldsymbol{x} \in \overline{B(\boldsymbol{x}^*, r)} \setminus \{\boldsymbol{x}^*\}$. If $0 < \eta < 2/\lambda_n$, where $\lambda_n$ is the largest eigenvalue of $\nabla^2 f(\boldsymbol{x}^*)$, then there exists an open set $U \subset B(\boldsymbol{x}^*, r)$ containing $\boldsymbol{x}^*$ such that the following forward-invariance property holds,

$$\boldsymbol{x} \in U \implies F(\boldsymbol{x}) = \boldsymbol{x} - \eta\nabla f(\boldsymbol{x}) \in U. \tag{8}$$

Here, $B(\boldsymbol{x}^*, r)$ is the open ball centered at $\boldsymbol{x}^*$ with radius $r$, and $\overline{B(\boldsymbol{x}^*, r)}$ is its closure. The foward-invariance is one of the most important properties for GD, which states that once the GD trajectory falls in to the forward-invariance set $U$, it will never escape from it.

Noticed that Lemma 1 remains valid for all $0 < \eta < 2/\lambda_n$, and it will be utilized in the proof for both cases of small or large learning rates. Its proof is detailed in Appendix B. Next, leveraging $\Omega$ and $C$ from Lemma 1, we select a constant $\varepsilon > 0$ satisfying

$$B(\boldsymbol{0}, \varepsilon) \subset \Omega, \quad C\varepsilon + (a + nC\varepsilon)^2 < a, \quad \varepsilon \leq \frac{a - b}{3n^2 C}. \tag{9}$$

With this choice, we first prove that the Theorem 1 holds for almost all initial points $\boldsymbol{x}_0 \in B(\boldsymbol{0}, \varepsilon)$. We now provide the following lemma.

**Lemma 2**: Consider the loss function $f$ defined in Theorem 1 with learning rate $0 < \eta < 2/(\lambda_1 + \lambda_n)$. For the constant $\varepsilon$ defined in equation 9, the following forward-invariance properties holds for $k \geq 0$,

$$\boldsymbol{x}_0 \in B(\boldsymbol{0}, \varepsilon) \implies \boldsymbol{x}_k \in B(\boldsymbol{0}, \varepsilon). \tag{10}$$

By this lemma, we see that $B(\boldsymbol{0}, \varepsilon)$ is also a forward invariant set, i.e., the GD trajectories dived into $B(\boldsymbol{0}, \varepsilon)$ remain confined within $B(\boldsymbol{0}, \varepsilon)$. The detailed proof of Lemma 2 is provided in Appendix C.

Having established these lemmas, we now prove Theorem 1 for the case of small learning rate $0 < \eta < 2/(\lambda_1 + \lambda_n)$. Firstly, define

$$S = \left\{\boldsymbol{x}_0 \in B(\boldsymbol{0}, \varepsilon) : \forall k \geq 0, |x_{k,1}| < \sum_{i=2}^{n} |x_{k,i}|\right\}. \tag{11}$$

One can prove that $S$ has zero measure (the detailed proof of this technical issue is provided in Appendix D). Then we only need to consider $\boldsymbol{x}_0 \in B(\boldsymbol{0}, \varepsilon) \setminus S$. Specifically, by definition of $S$ in equation 11, we only need to consider the initial point $\boldsymbol{x}_0 \in B(\boldsymbol{0}, \varepsilon)$ such that there exists $k^* \geq 0$,

$$|x_{k^*,1}| \geq \sum_{i=2}^{n} |x_{k^*,i}|. \tag{12}$$

In this case, it's easy to see that $|x_{k^*,1}| \geq |x_{k^*,i}|$ for $2 \leq i \leq n$. Thus, by Cauchy's inequality, $\|\boldsymbol{x}_{k^*}\|^2 \leq nx_{k^*,1}^2$. Then, according to equation 6 and Lemma 1,

$$|x_{k^*+1,1}| \geq a|x_{k^*,1}| - |g_{k^*,1}| \geq a|x_{k^*,1}| - C\|\boldsymbol{x}_{k^*}\|^2 \geq a|x_{k^*,1}| - nCx_{k^*,1}^2. \tag{13}$$

Moreover, for $2 \leq i \leq n$, also by equation 6 and Lemma 1,

$$|x_{k^*+1,i}| \leq |1 - \eta\lambda_i| \, |x_{k^*,i}| + |g_{k^*,i}| \leq b \, |x_{k^*,i}| + C \, \|\boldsymbol{x}_{k^*}\|^2 \leq b \, |x_{k^*,i}| + nC x_{k^*,1}^2. \tag{14}$$

It follows that

$$\sum_{i=2}^{n} |x_{k^*+1,i}| \leq b \sum_{i=2}^{n} |x_{k^*,i}| + n(n-1) C x_{k^*,1}^2. \tag{15}$$

Based on equation 12 and equation 15, we have,

$$\sum_{i=2}^{n} |x_{k^*+1,i}| \leq b \, |x_{k^*,1}| + n(n-1) C x_{k^*,1}^2. \tag{16}$$

Then, by equation 9, equation 13 and equation 16, with $|x_{k^*,1}| \leq \|\boldsymbol{x}_{k^*}\| < \varepsilon$,

$$|x_{k^*+1,1}| \geq \sum_{i=2}^{n} |x_{k^*+1,i}| \, . \tag{17}$$

In this way, by Lemma 2, as $\|\boldsymbol{x}_0\| < \varepsilon$, we know that $\|\boldsymbol{x}_k\| < \varepsilon$ holds for any $k \geq 0$. Then, by induction, we conclude that for all $k \geq k^*$,

$$|x_{k,1}| \geq \sum_{i=2}^{n} |x_{k,i}| \, . \tag{18}$$

Moreover, to ensure our conclusion, we have to show if $x_{k^*,1} > 0$, then for all $k \geq k^*$, $x_{k,1} > 0$. Actually, if $x_{k^*,1} > 0$, then by equation 6, equation 9 and the same induction in equation 13,

$$x_{k+1,1} \geq ax_{k,1} - nC x_{k,1}^2 \geq (a - nC\varepsilon) \, x_{k,1} > 0. \tag{19}$$

The above claim can be then proved inductively. In the same way, if $x_{k^*,1} < 0$, then for all $k \geq k^*$, $x_{k,1} < 0$. In this situation, we know that $x_{k,1} \neq 0$ for all $k \geq k^*$, if $x_{k^*,1} \neq 0$. Furthermore, with equation 18, based on the same induction of equation 13 and $|x_{k,1}| \leq \varepsilon$, we have, for all $k \geq k^*$,

$$|x_{k+1,1}| \geq a \, |x_{k,1}| - nC x_{k,1}^2 \geq (a - nC\varepsilon) \, |x_{k,1}| \, . \tag{20}$$

Next, similarly, with equation 18, based on the same induction of equation 14, for $k \geq k^*$ and $2 \leq i \leq n$, we have,

$$|x_{k+1,i}| \leq b \, |x_{k,i}| + C \, \|\boldsymbol{x}_k\|^2 \leq b \, |x_{k,i}| + nC x_{k,1}^2. \tag{21}$$

Since $\varepsilon$ satisfies equation 9, then $a - nC\varepsilon > b$, one can pick $\alpha$ satisfying $1 < \alpha < 2$ such that $q = b/(a - nC\varepsilon)^\alpha < 1$. Then, according to equation 20 and equation 21, for $k \geq k^*$ and $2 \leq i \leq n$, we have,

$$\frac{|x_{k+1,i}|}{|x_{k+1,1}|^\alpha} \leq \frac{b \, |x_{k,i}|}{(a - nC\varepsilon)^\alpha \, |x_{k,1}|^\alpha} + \frac{nC \, |x_{k,1}|^2}{(a - nC\varepsilon)^\alpha \, |x_{k,1}|^\alpha} = q \frac{|x_{k,i}|}{|x_{k,1}|^\alpha} + \frac{nC |x_{k,1}|^{2-\alpha}}{(a - nC\varepsilon)^\alpha}. \tag{22}$$

Since $|x_{k,1}|^{2-\alpha} < 1$, equation 22 can be reduced to

$$\frac{|x_{k+1,i}|}{|x_{k+1,1}|^\alpha} \leq q \frac{|x_{k,i}|}{|x_{k,1}|^\alpha} + \frac{nC}{(a - nC\varepsilon)^\alpha}. \tag{23}$$

Still, by induction, equation 23 implies that, for all $k \geq k^*$ and $2 \leq i \leq n$,

$$\frac{|x_{k,i}|}{|x_{k,1}|^\alpha} \leq q^{k-k^*} \frac{|x_{k^*,i}|}{|x_{k^*,1}|^\alpha} + \frac{nC}{(a - nC\varepsilon)^\alpha} \sum_{j=0}^{k-k^*} q^j. \tag{24}$$

As $\sum_{j=0}^{k-k^*} q^j < 1/(1-q)$, we have, for all $k \geq k^*$ and $2 \leq i \leq n$,

$$\frac{|x_{k,i}|}{|x_{k,1}|^\alpha} \leq q^{k-k^*} \frac{|x_{k^*,i}|}{|x_{k^*,1}|^\alpha} + \frac{nC}{(1-q)(a - nC\varepsilon)^\alpha}. \tag{25}$$

Hence, there exists a constant $C_1 > 0$ such that the following estimation holds for $k \geq k^*$ and $2 \leq i \leq n$, $\frac{|x_{k,i}|}{|x_{k,1}|^\alpha} \leq C_1$. As a result, for all $2 \leq i \leq n$, $\lim_{k \to \infty} \left| \frac{x_{k,i}}{x_{k,1}} \right| = 0$. Therefore, based on the above induction, on one hand, if $\boldsymbol{x}_{k^*,1} \neq 0$,

$$\lim_{k \to \infty} \frac{\boldsymbol{x}_k}{\|\boldsymbol{x}_k\|} = (\operatorname{sign}(x_{k^*,1}), 0, \cdots, 0). \tag{26}$$

This situation corresponds to the case where the GD trajectory eventually becomes dominated by the direction of the eigenvector corresponding to the smallest eigenvalue of $\nabla^2 f(\boldsymbol{x}^*)$. This behavior is intuitive and expected for almost all initial points in $B(\boldsymbol{0}, \varepsilon)$.

On the other hand, if $x_{k^*,1} = 0$, we see that $\boldsymbol{x}_{k^*} = \boldsymbol{0}$ since $\boldsymbol{x}_0 \notin S$, meaning that the GD trajectory reached $\boldsymbol{x}^* = \boldsymbol{0}$ after only finite iterations. Define

$$W_k = \{\boldsymbol{x}_0 \in B(\boldsymbol{0}, \varepsilon) : \boldsymbol{x}_k = \boldsymbol{0}\}, \tag{27}$$

then we have $\boldsymbol{x}_0 \in W_{k^*}$. According to the assumption on the GD dynamics in Theorem 1, $W_k$ is a zero measure set since $W_k = F^{-k}(\{\boldsymbol{0}\})$), and thus the set $S \cup (\cup_{k=0}^\infty W_k)$ has zero measure. In this way, for each initial point $\boldsymbol{x}_0 \in B(\boldsymbol{0}, \varepsilon) \setminus (S \cup (\cup_{k=0}^\infty W_k))$, according to equation 26, the convergence direction of GD exists. Finally, return to the set $V_\eta$, define

$$\widetilde{W}_k = \{\boldsymbol{x}_0 \in V_\eta : \boldsymbol{x}_k = \boldsymbol{0} \text{ or } \boldsymbol{x}_k \in S\}. \tag{28}$$

It is clear that the measure of $\widetilde{W}_k$ is zero, thus $\cup_{k \geq 0} \widetilde{W}_k$ also has zero measure. Then for each initial point $\boldsymbol{x}_0 \in V_\eta \setminus (\cup_{k \geq 0} \widetilde{W}_k)$, as its GD trajectories will finally enter the set $B(\boldsymbol{0}, \varepsilon) \setminus (S \cup (\cup_{k=0}^\infty W_k))$, the theorem is finally proved.

The detailed proof of Theorem 1 with large learning rate is provided in Appendix E.

## 3.2 SHARPNESS OSCILLATION AND DISCUSSION OF EOS

Consider a function $f : \mathbb{R}^n \to \mathbb{R}$ satisfying the assumptions of Theorem 1 with a local minimum $\boldsymbol{x}^* = \boldsymbol{0}$. Suppose that the Hessian $\nabla^2 f(\boldsymbol{0})$ have distinct eigenvalues satisfying $0 < \lambda_1 < \lambda_2 < \cdots < \lambda_n$. By classical matrix perturbation theory (Kato, 1995; Horn & Johnson, 2013; Bhatia, 2013), we know that there exists a neighborhood $U$ of $\boldsymbol{x}^*$ such that the eigenvalues $\lambda_1(\boldsymbol{x}), \ldots, \lambda_n(\boldsymbol{x})$ of $\nabla^2 f(\boldsymbol{x})$ are differentiable functions of $\boldsymbol{x}$, and $\lambda_1(\boldsymbol{x}) < \ldots < \lambda_n(\boldsymbol{x})$. Then, by Taylor expansion, we have, for the maximal eigenvalue $\lambda_n(\boldsymbol{x})$ and $\boldsymbol{x} \in U$,

$$\lambda_n(\boldsymbol{x}) = \lambda_n(\boldsymbol{0}) + \boldsymbol{\omega}^\top \boldsymbol{x} + o(\|\boldsymbol{x}\|), \tag{29}$$

in which, by definition, $\lambda_n(\boldsymbol{0}) = \lambda_n$ is the maximal eigenvalue at $\boldsymbol{0}$ and $\boldsymbol{\omega} = \lambda_n'(\boldsymbol{0})$. Accordingly, by Theorem 1, the GD trajectory $\{\boldsymbol{x}_k\}_{k \geq 0}$ converges or alternatively converges along a certain direction $\boldsymbol{v} \in \mathbb{R}^n$. As a result, we may conclude that: For small learning rate $\eta \in (0, 2/(\lambda_1 + \lambda_n))$, there exists a constant $C_\eta$ such that

$$\lambda_n(\boldsymbol{x}_k) = \lambda_n + C_\eta \|\boldsymbol{x}_k\| + o(\|\boldsymbol{x}_k\|). \tag{30}$$

Therefore, ignoring the higher order term of $\|\boldsymbol{x}_k\|$, we have, $\lambda_n(\boldsymbol{x}_k) \downarrow \lambda_n$ or $\lambda_n(\boldsymbol{x}_k) \uparrow \lambda_n$ depending on the sign of $C_\eta$. For large learning rate $\eta \in (2/(\lambda_1 + \lambda_n), 2/\lambda_n)$, there exists a constant $C_\eta$ such that

$$\lambda_n(\boldsymbol{x}_k) = \lambda_n + (-1)^k C_\eta \|\boldsymbol{x}_k\| + o(\|\boldsymbol{x}_k\|). \tag{31}$$

Similarly, ignoring the higher order term of $\|\boldsymbol{x}_k\|$, we have, for instance, when $C_\eta > 0$, $\lambda_n(\boldsymbol{x}_k)$ will oscillate converge to $\lambda_n$, i.e., $\lambda_n(\boldsymbol{x}_{2k}) \downarrow \lambda_n$ and $\lambda_n(\boldsymbol{x}_{2k+1}) \uparrow \lambda_n$. Moreover, as $\lambda_n < 2/\eta < \lambda_1 + \lambda_n$, compared with the above case, the sharpness $\lambda_n(\boldsymbol{x}_{2k})$ is more likely larger than $2/\eta$ for the case of small $k$, i.e., for the first several $\boldsymbol{x}_k$ in GD trajectory. The above discussion may provide new insights to the sharpness fluctuation phenomenon of EoS.

We now present an example to illustrate our idea. Consider here a loss function of two variables, $f(x, y) = x^2 + y^2/2 + x^2 y + x^3$, which has a local minimum $(0, 0)$ with $\lambda_1 = 1$ and $\lambda_2 = 2$. The GD dynamics and sharpness evolution under different learning rates are shown in Figure 2. In Figure 2a and 2b, the learning rate is taken as $\eta = 0.1 < 2/(\lambda_1 + \lambda_2) = 2/3$. In these two figures, the GD trajectory converges to the minimum along $\boldsymbol{v} = (0, \mp 1)$, and the maximum eigenvalue $\lambda_2(\boldsymbol{x}_k)$ converges monotonically to $\lambda_2 = 2$, consistent with our theoretical prediction for small

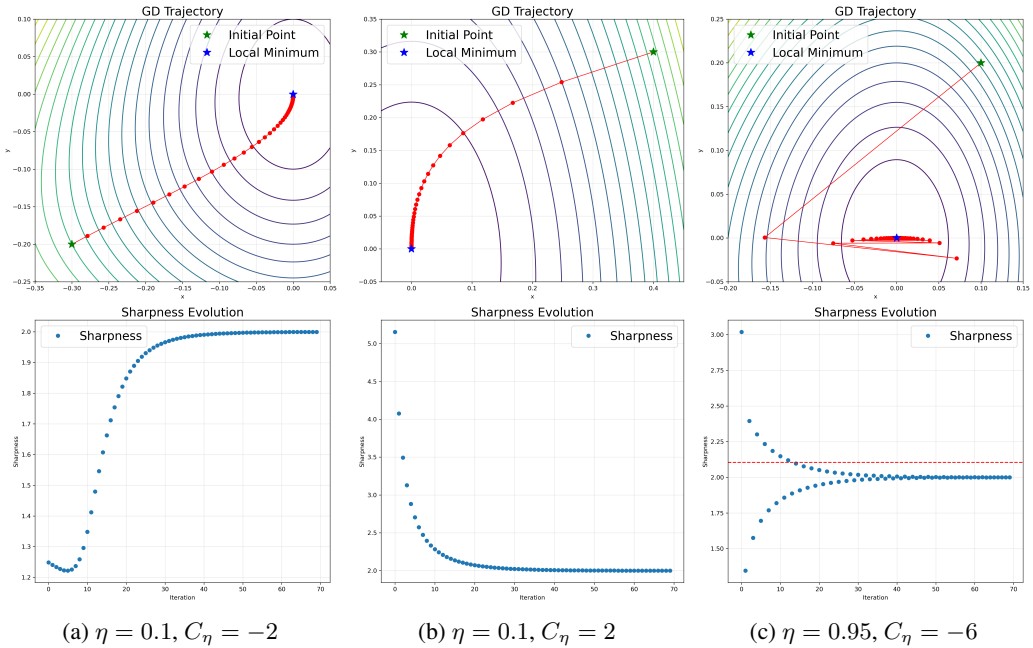

Figure 2: GD behavior on $f(x, y) = x^2 + y^2/2 + x^2 y + x^3$ under different learning rates. The top row shows the GD trajectories, and the bottom row shows the evolution of the sharpness along GD trajectory.

learning rates. On the other hand, in Figure 2c, the learning rate $\eta = 0.95$ satisfies $2/(\lambda_1 + \lambda_2) = 2/3 < \eta < 2/\lambda_2 = 1$. As a result, the GD trajectory exhibits oscillations along the sharpest direction $\boldsymbol{v} = (1, 0)$. Correspondingly, the maximum eigenvalue of the Hessian oscillates around the asymptotic value $\lambda_2$, and may occasionally exceed the theoretical upper bound $2/\eta \approx 2.105$, which is shown as the dashed red line in this figure. This fluctuation is a manifestation of the EoS phenomenon, but the long-term convergence of the oscillation envelope toward $\lambda_2$ supports the conclusion of our theorem. More results about this experiment are provided in Appendix F.

### 3.3 CONVERGENCE DIRECTIONS FOR MODERN OPTIMIZERS

Having rigorously established in Theorem 1 that GD trajectories admit a convergence direction, we are curious whether this phenomenon might extend to more widely used optimization algorithms in deep learning. Surprisingly, our experiments reveal that the existence of convergence directions is not confined to the deterministic and idealized setting of vanilla GD. Similar behaviors also emerge when using SGD with momentum and Adam in multi-class classification tasks.

To explore this, we conduct experiments on the CIFAR-10 dataset, using a convolutional neural network architecture (the details are provided in Appendix G). We train this network using two widely adopted optimization methods: SGD with momentum and Adam. For each optimizer, we monitor both the training loss and the directional alignment of successive update steps throughout training. In particular, we track the cosine similarity between consecutive GD parameter updates, defined as,

$$\cos \langle \Delta \boldsymbol{x}_{k+1}, \Delta \boldsymbol{x}_k \rangle = \frac{\langle \Delta \boldsymbol{x}_k, \Delta \boldsymbol{x}_{k+1} \rangle}{\|\Delta \boldsymbol{x}_k\| \|\Delta \boldsymbol{x}_{k+1}\|}, \tag{32}$$

where $\Delta \boldsymbol{x}_k = \boldsymbol{x}_{k+1} - \boldsymbol{x}_k$, which captures whether the optimizer's trajectory aligns with a stable direction over time. The results are shown in Figure 3. For all optimization methods, $\cos \langle \Delta \boldsymbol{x}_{k+1}, \Delta \boldsymbol{x}_k \rangle$ tends to 1 when the loss function converges stably in descending manner, which indicates the convergence direction emerges and supports our claim.

This finding has important implications for our understanding of optimization dynamics. It suggests that the asymptotic alignment behavior may reveal several promising research directions. The consistency of convergence directions across algorithms could inform better learning rate schedules,

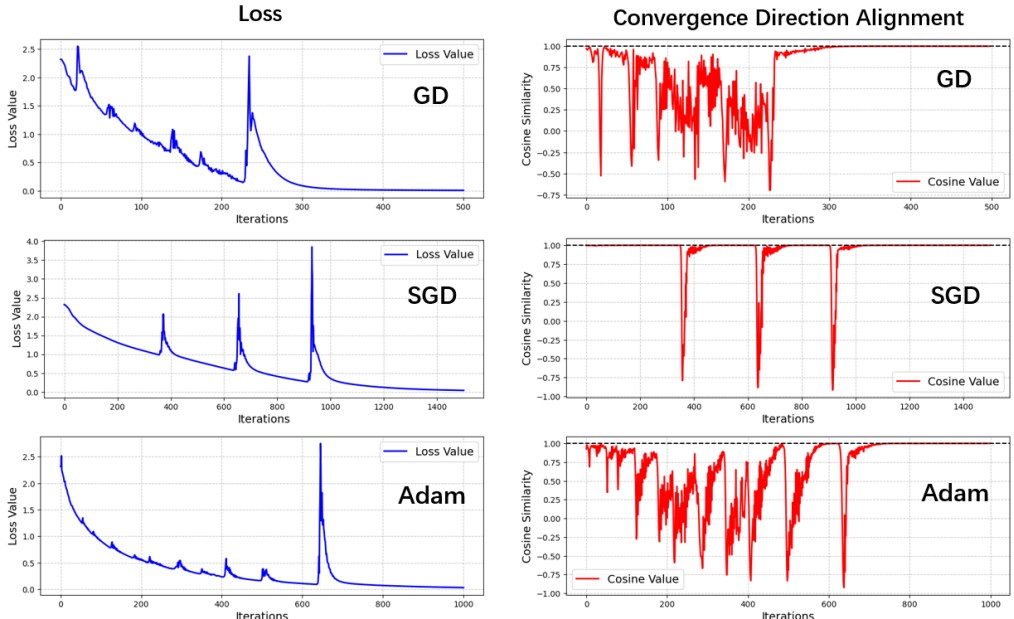

Figure 3: Convergence directions for GD, SGD, and Adam. Each row presents the training loss (left) and the convergence direction alignment (right) for a different optimizer.

initializations, or optimizers. Understanding their emergence, especially in adaptive methods, may offer insights into convergence and generalization.

## 4    CONCLUSION

In this paper, we investigate the asymptotic directional behavior of GD and establish new theoretical results. Motivated by the Gradient Conjecture from continuous-time gradient flow, we prove that under mild assumptions, the discrete-time GD also admits a similar directional property. Moreover, we discuss the oscillating behavior of sharpness by deriving the variation of eigenvalues near the minimum. Furthermore, we also find that this result holds for SGD and Adam.

This work opens several promising directions for future research.

- We only consider the locally strong convex function in the proposed theorem. It is important to investigate whether directional convergence can be extended to more general functions and other optimization algorithms, such as momentum-based or adaptive methods.

- A quantitative study of convergence speed in the direction could offer sharper theoretical guarantees and practical insights. Such results may inform the design of new optimization algorithms that leverage directional behavior for improved efficiency or generalization. Understanding these aspects could help unify optimization methods under a broader theoretical framework.

- An intriguing future direction is to explore whether the phenomenon of convergence directions observed in GD is further amplified or structurally modified under the use of periodically long steps, as suggested by the non-monotonic but globally contracting patterns in Grimmer (2024).

- Building upon the  Kurdyka-Łojasiewicz  (KL) inequality based works proposed in Qiu et al. (2024); Attouch et al. (2010), a promising direction is to extend the analysis to functions defined with the KL condition. Another interesting line of work is to investigate the finite-time behavior and generalization performance of stochastic optimization under the approximate descent paradigm. Finally, integrating the KL inequality with data-dependent sampling strategies or overparameterized models could provide new insights into convergence dynamics in modern machine learning.

ACKNOWLEDGEMENTS

This work was supported by the National Natural Science Foundation of China under Grants 62372037 and U24B20179.

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
