# OpenReview forum: "On the Convergence Direction of Gradient Descent"
_ICLR.cc/2026/Conference — ICLR 2026 Poster_

### Official Review · Reviewer_YPEn · 2025-10-29

**Soundness:** 1
**Presentation:** 3
**Contribution:** 2
**Rating:** 2
**Confidence:** 3

**Summary:**

Extending to the discrete setting a result known for the gradient flow, the authors characterize how gradient descent, assuming it converges to some local minima, will converge to this limit. They show that, depending on stepsize, it will either converge along a fixed direction, or oscillate along it. Numerical experiments are provided on toy examples to illustrate this result, and also to observe empirically, on a training neural network task, how behave the angle between consecutive iterates for several optimizers. A section is devoted to the connection of their work with the Edge of Stability (EoS) phenomenon.

**Strengths:**

The presentation is clear, the paper well organized. I think not assuming global smoothness a priori and instead characterizing the convergence according to the eigenvalue of the Hessian at a local minima is an interesting viewpoint, such that the main result (Theorem 1) is interesting.

**Weaknesses:**

I express severe doubts about the relevance of the connection established between the author's work and the EoS phenomenon, which is presented as a central contribution.


- In their main result, the authors assume the function is $C^3$, and that gradient descent converges to a strict local minimizer. Because of the $C^3$ assumption, there exists a vicinity of this local minimizer such that the function is strongly convex and $L$-smooth for some $L$, with $L$ as close as we want to $\lambda_n$.
As gradient descent is assumed to converge to this local minimizer, the algorithm eventually enters this vicinity, and stay inside it. With this regard:

     (i) It is implicitly assumed that asymptotically, the algorithm lies in a strongly convex, $L$-smooth landscape, with $\eta < \frac{2}{L}$.

     (ii) In this setting, even if the sharpness exhibit oscillations , the loss still decreases monotonically, as it seems to be observed on figure 2.c. This is not consistent with the typical behavior of the EoS regime, where the loss is not monotonically decreasing, but only decreases in the long run.

     (iii) According to [1], a quadratic growth property does not fit the local behavior of neural network losses around minimizers, whose growth is more likely to be sub linear. Such a setting is not covered by the implicit "local strong convexity" assumption of the authors.

For those reasons, it is not clear to me whether this work improves or not our understanding of the EoS phenomenon. I believe the claimed connection should be substantiated more clearly.

[1] Helusive traductionao Ma, Lei Wu, and Lexing Ying. The multiscale structure of neural network loss functions: The
effect on optimization and origin

Minor remarks
- The labels on figure 1 are very small

**Questions:**

- On figure 2)a), as the convergence occurs along the y-axis direction, why the sharpness does not converge to 1 instead of 2 ?

- Under equation (5), the justification that $a > b $ seems evasive to me. Could you elaborate ?

---

> ### Author Response · Authors · 2025-11-26
> **Rebuttal for Comments from Reviewer YPEn**
>
> >I express severe doubts about the relevance of the connection established between the author's work and the EoS phenomenon, which is presented as a central contribution.
>
> We would like to clarify that our primary contribution is not to provide a new characterization or explanation of the EoS phenomenon. The central focus of our work is the rigorous theoretical proof of the existence of a limit (or convergence) direction for the trajectories of GD dynamics. The connection to EoS is mainly on the stage that GD is going to converge. It should be considered as a possible implication or interpretation of our theoretical results, rather than as our main contribution.
>
> We acknowledge the reviewer's point that the assumption of a $C^3$ function and convergence to a strict local minimizer ensures that, in a neighborhood of the minimizer, the function is locally strongly convex and $L$-smooth. However, our results and analysis do not depend on globally enforcing a quadratic or strongly convex structure. The local analysis is introduced solely to enable a rigorous characterization of asymptotic trajectories and to establish the existence of the convergence direction.
>
> Regarding the reviewer's point that such an assumption differs from the non-monotonic loss behavior typical in EoS regimes, we note that our experiments in Figure 2(c) indeed show oscillatory behavior in the sharpness. We apologize if the visual presentation led to confusion --- the oscillations are present but the connecting lines between discrete samples were omitted in the plot, which may have made this less apparent. To eliminate such misunderstanding, we upload a new version of Figure 2(c) in the supplementary material.
>
> As for the relation to subquadratic growth: We appreciate the reviewer drawing attention to this work. However, the analysis in Ma et al. primarily studies a non-local region of loss landscape, i.e., the landscape of loss function $L(x)$ as $x \to \infty$. On the contrast, our focus is the convergence direction as GD dynamics approaches to the local minimum.
>
> >On figure 2(a), as the convergence occurs along the y-axis direction, why the sharpness does not converge to 1 instead of 2 ?
>
> We appreciate this question and would like to clarify. The "sharpness" refers to the largest eigenvalue of the Hessian evaluated along the GD trajectory. As the trajectory converges toward a local minimizer, the direction of motion becomes increasingly aligned with the eigenvector corresponding to the smallest eigenvalue when the learning rate is small. Therefore, even though the trajectory converges primarily along the $y$-axis (corresponding to the smaller eigenvalue), the reported sharpness value approaches the larger eigenvalue (approximately 2) rather than 1. This is consistent with the definition and theoretical expectations.
>
> >Under equation (5), the justification that $a>b$ seems evasive to me. Could you elaborate?
>
> We appreciate the reviewer's suggestion. In fact, when $0 < \eta < \frac{2}{\lambda_1 + \lambda_n}$, we have $1-\eta\lambda_1>0$, and, for $i>1$,
> $$1 - \eta \lambda_1 > \pm (1-\eta \lambda_i).$$
> Therefore,
> $$\max_{1 \le i \le n} | 1 - \eta \lambda_i | = 1 - \eta \lambda_1,$$
> Under the definitions of $a$ and $b$ used in the equation (5) of main text, this implies $a > b$.

---

> > ### Comment · Reviewer_YPEn · 2025-11-27
> >
> > Thank you for the answer and clarifications.
> > "We would like to clarify that our primary contribution is not to provide a new characterization or explanation of the EoS phenomenon."
> > This is not apparent when reading the manuscript. In the abstract itself: "our result sheds light on the phenomenon of EoS, explaining why sharpness oscillates even as the loss converges".
> > I still do not see how Figure 2(c) relates to the EoS phenomenom as it appear in applications. I believe the justifications of the relation between this phenomenom and your result are still not convincing, so I will keep my current score.

---

> > > ### Author Response · Authors · 2025-11-30
> > > **Rebuttal for New Comments from Reviewer YPEn**
> > >
> > > We thank the reviewer for the comments and we again clarify our contribution and its relation to the EoS phenomenon.
> > >
> > > **Clarifying our statement in the abstract**
> > >
> > >    We acknowledge the reviewer’s concern in the abstract:
> > >    “our result sheds light on the phenomenon of EoS, explaining why sharpness oscillates even as the loss converges.”
> > >    Our intention was not to claim that the proposed analysis provides a full characterization or a definitive explanation of the EoS phenomenon. Rather, our theoretical results rigorously identify a property of GD.
> > >
> > > **Clarification regarding Figure 2(c)**
> > >
> > >    Figure 2(c) is intended to illustrate two qualitatively different stages of sharpness evolution under GD on one toy example.
> > >
> > >    * In the early iterations, the sharpness oscillates around approximately $2/\eta$, which corresponds to the empirically observed EoS phenomenon.
> > >    * In later iterations, as GD approaching to a local minimum, the sharpness continues to oscillate but gradually converges to the sharpness of the minimizer, which is distinct from the EoS value.
> > >
> > >    The purpose of this figure is therefore not to present an empirical demonstration of EoS in practical deep learning, but to highlight that our theoretical mechanism produces both (i) an initial EoS-like oscillatory regime and (ii) a subsequent oscillatory-but-convergent regime, the latter being the focus of our theoretical analysis.
> > >
> > > **Motivation and scope of the work**
> > >
> > >    Our work is motivated by EoS and the gradient conjecture of René Thom (see Kurdyka et al., "Proof of the gradient conjecture of R. Thom", Annals of Mathematics), which concerns intrinsic properties of gradient flows. Our main objective is to provide a rigorous analysis of an intrinsic behavior of GD near the minimum. Our results show that GD with small learning rates are similar to the case of gradient flow, while large learning rates are essentially different from that of gradient flow. In this sense, the primary contribution lies in theoretical understanding rather than in providing a comprehensive explanation of EoS.
> > >
> > >   While we explore connections to the EoS phenomenon and discuss how our findings offer a new theoretical insight related to sharpness oscillations, this is not the central goal of the paper.

---

### Official Review · Reviewer_Cqt9 · 2025-10-30

**Soundness:** 4
**Presentation:** 3
**Contribution:** 3
**Rating:** 8
**Confidence:** 4

**Summary:**

This paper studies the asymptotic behavior of gradient descent as it converges to a minimizer, on generic objective functions. The main assumption is that the objective is strongly convex around the minimizer (i.e. the Hessian at the minimizer has no zero eigenvalues).   The paper proves that there exactly two possible modes of convergence: either gradient descent converges along a specific direction (i.e. cosine distance between successive iterates is $\approx 1$), or it oscillates along a particular line (i.e. cosine distance between successive iterates is $\approx -1$).   They prove that the first case happens when $0 < \eta < 2 / (\lambda_1 + \lambda_n)$ and that the second case happens when $2 / (\lambda_1 + \lambda_n) < \eta < 2 / \lambda_n$, where $\lambda_1$ and $\lambda_n$ are the largest and smallest Hessian eigenvalues at the minimizer.

I think the basic intuition comes from the case of a quadratic objective.  Here, the rate of convergence along each eigenvector direction is $1 - \eta \lambda_i$, where $\lambda_i$ is the eigenvalue.  Asymptotically, we will be converging along the eigenvector for which $|1 - \eta \lambda_i|$ is largest.  There are two possible cases:
  - if $\eta < 2 / (\lambda_1 + \lambda_n)$, then this direction will be the eigenvector corresponding to the _smallest_ eigenvalue $\lambda_n$ and the contraction factor $1 - \eta \lambda_n$ will be positive and we will converge along that particular direction
  - if $\eta > 2 / (\lambda_1 + \lambda_n)$, then this direction will be the eigenvector corresponding to the _largest_ eigenvalue $\lambda_1$, and the contraction factor $1 - \eta \lambda_1$ will be negative and we will oscillate along that line.

The paper basically makes this intuition precise for general objectives.

**Strengths:**

The paper nicely tackles an interesting problem in a fully rigorous manner.

**Weaknesses:**

The asymptotic behavior of an optimizer near a minimum may not be that relevant for practical deep learning tasks, where practitioners generally care about the behavior of the optimizer far from any minimum.

Another weakness is that the paper only studies the case where gradient descent converges to a point, whereas it is also possible for gradient descent to wind up in a orbit where it oscillates indefinitely, for example see [Chen and Bruna '23] or [Ghosh et al '25].

Lei Chen and Joan Bruna.  "Beyond the edge of stability via two-step gradient updates."  ICML '23.

Avrajit Ghosh, Soo Min Kwon, Rongrong Wang, Saiprasad Ravishankar and Qing Qu.  "Learning Dynamics of Deep Linear Networks Beyond the Edge of Stability."  ICLR '25.

**Questions:**

- What happens when the learning rate is greater than $2/\lambda_n$ -- can you prove that convergence is impossible?
- For the experiments in section 3.3, I think you used cross-entropy loss.  For cross-entropy loss, optimizers generally leave EOS before the end of training (see Cohen et al '21, Appendix C), which is what we see here.  I think that if you used MSE loss, you might see the other, oscillatory convergence regime.

---

> ### Author Response · Authors · 2025-11-26
> **Rebuttal for Comments from Reviewer Cqt9**
>
> >What happens when the learning rate is greater than $2/\lambda_{\max}$ --- can you prove that convergence is impossible?
>
> We thank the reviewer for raising this point. When the learning rate exceeds the critical threshold $2 / \lambda_{\max}$, where $\lambda_{\max}$ is the largest eigenvalue of the Hessian at the minimum, divergence can indeed be shown rigorously. Specifically, excluding a zero-measure set of initializations, convergence becomes impossible in this regime (see e.g., Theorem 1, Ahn et al. ``Understanding the Unstable Convergence of Gradient Descent'' ICML'22).
>
> >For the experiments in Section 3.3, I think you used cross-entropy loss. For cross-entropy loss, optimizers generally leave the edge-of-stability (EOS) regime before the end of training (see cohen2021edge, Appendix~C) , which is what we see here. I think that if you used MSE loss, you might see the other, oscillatory convergence regime.
>
> We thank the reviewer for this valuable observation. Yes, our experiments employ the cross-entropy loss. We agree that using MSE loss may highlight a distinct oscillatory convergence regime, consistent with our theoretical framework. Also, we upload new experiment using MSE loss on same task in the new supplementary materials.

---

### Official Review · Reviewer_dfgz · 2025-10-31

**Soundness:** 4
**Presentation:** 2
**Contribution:** 3
**Rating:** 6
**Confidence:** 3

**Summary:**

This paper study the convergence in direction of the iterates of gradient descent. The main results of the papers identifies two regimes. If the step size $\eta$ verifies $0<\eta<\frac2{\lambda_1+\lambda_n}$ where $\lambda_1$ and $\lambda_n$ are respectively the smallest and the largest eigenvalues of the Hessian of the objective function $f$ at a critical point, the iterates of gradient descent converge in direction. If $\frac2{\lambda_1+\lambda_n}<\eta<\frac2{\lambda_n}$, the iterates exhibit an alternating convergence direction. The paper then discusses the implications of these results for the edge of stability and provides numerical experiments suggesting that the theoretical findings extend to other optimizer such as SGD with momentum and Adam.

**Strengths:**

* **S1**: The authors establish convergence direction of gradient descent. The result is sound and novel to my knowledge.

* **S2**: The established results provide insight on the behavior of the sharpness during the optimization trajectory.

* **S3**: Numerical experiments suggest that the theoretical results established for GD may extend to SGD with momentum and Adam.

**Weaknesses:**

* **W1**: The experiments appear to be based on a single run. Since some of the compared algorithms are stochastic, multiple runs should be conducted. The results should then report the average or median cosine metric along with standard deviations or quantiles to reflect variability and ensure statistical reliability.

* **W2**: The code of the experiments is not provided, hindering reproducibility.

**Questions:**

* **Q1**: In the experiments with neural networks, how do the learning rates compare with the sharpness? It seems that only the convergence direction occurs. Do SGD and Adam exhibit alternative convergence direction with higher step sizes?

---

> ### Author Response · Authors · 2025-11-26
> **Rebuttal for Comments from Reviewer dfgz**
>
> >W1: The experiments appear to be based on a single run. Since some of the compared algorithms are stochastic, multiple runs should be conducted. The results should then report the average or median cosine metric along with standard deviations or quantiles to reflect variability and ensure statistical reliability.
>
> >W2: The code of the experiments is not provided, hindering reproducibility.
>
> We thank the reviewer for these constructive comments. Actually, our experiments are not meant to provide new empirical findings, but to illustrate the theory. Our main contribution is to prove the existence of the limit direction for GD. Moreover, in the rebuttal period, we have conducted additional parallel experiments under the same experimental setup as in the main text. These new results are included in the supplementary materials. We also upload an example code in the supplementary materials.
>
> > In the experiments with neural networks, how do the learning rates compare with the sharpness? It seems that only the convergence direction occurs. Do SGD and Adam exhibit alternative convergence direction with higher step sizes?
>
> We appreciate the reviewer’s insightful question. Given that empirical evidence for the relationship between learning rate and sharpness has already been extensively investigated, e.g., in the work of Cohen et al., thus such numerical results are not included in our paper. Regarding the behavior of SGD and Adam, we observed that SGD can exhibit alternating convergence directions under certain settings; these examples are now included in the supplementary materials. For Adam, finding comparable cases is more subtle because of its momentum structure. Within our computational budget and the range of learning-rate configurations we explored, we did not observe analogous oscillatory behavior.

---

### Official Review · Reviewer_i7ue · 2025-11-01

**Soundness:** 3
**Presentation:** 3
**Contribution:** 3
**Rating:** 6
**Confidence:** 3

**Summary:**

The paper studies the asymptotic direction of gradient descent (GD) near an isolated local minimum. It proves that, if the GD sequence converges, then its normalized iterates converge either to a fixed direction (for smaller learning rates) or alternate between two opposite directions along a line (for larger but still stable learning rates). The boundary between these regimes is $2/(\lambda_1+\lambda_n)$, where $\lambda_1$ and $\lambda_n$ are the smallest and largest eigenvalues of $\nabla^2f(x^*)$. The authors connect this to the Edge of Stability (EoS) by showing that, via a first-order expansion of $\lambda_{\max}(\nabla^2f(x_k))$, sharpness can oscillate even as the loss converges. Empirically, toy 2D examples illustrate the two regimes, and preliminary experiments suggest similar directional alignment behavior for SGD and Adam on CIFAR-10 via cosine similarity of successive updates.

**Strengths:**

The paper pinpoints a clean, eigenvalue-driven dichotomy for gradient descent near an isolated local minimum: when $0<\eta<2/(\lambda_1+\lambda_n)$, the normalized iterates converge to a fixed direction, whereas for $2/(\lambda_1+\lambda_n)<\eta<2/\lambda_n$ they alternate between two opposite directions. This result is stated cleanly (Theorem 1) with transparent assumptions and offers an interpretable link to the “edge-of-stability” phenomenon.

The analysis is technically careful: the argument builds an invariant neighborhood around the minimizer, bounds higher-order terms so the smallest-eigenvalue component dominates for sufficiently small $\eta$, and identifies a measure-zero exceptional set of initialization.

The exposition is clear and visual: Sec. 3.2 connects directional behavior to sharpness oscillations via a local expansion of the largest Hessian eigenvalue, and Figs. 1 and 2 make the two regimes concrete on toy problems. Although preliminary, the empirical section suggests similar directional alignment patterns for SGD and Adam on CIFAR-10, hinting at broader relevance beyond plain GD.

To sum up:
1. The paper has strong mathematical analysis with clear assumptions, statements and proofs.
2. The work includes experimental work that compares the proposed method with other works.
3. The writing is generally clear with nice flow.

**Weaknesses:**

My only concern is that the empirical validation is limited and largely qualitative: it relies on toy 2D landscapes and a single CIFAR-10 configuration, without uncertainty quantification, ablations, or explicit tracking of the normalized iterate direction toward a limiting vector/pair. Strengthening the evidence with additional toy problems and a broader deep-learning suite, e.g., multiple architectures on CIFAR-10 and CIFAR-100, would make the claims more compelling.

**Questions:**

1. Can you include more experiments as discussed in Weaknesses?

---

> ### Author Response · Authors · 2025-11-26
> **Rebuttal for Comments from Reviewer i7ue**
>
> We thank the reviewer for this helpful suggestion. Actually, these empirical studies are not the contribution of the paper, but the theoretical result of Theorem 1, which rigorously prove the existence for limit directions for GD.
> To illustrate that our observations are not specific to a particular architecture, we have additionally conducted experiments on ResNet-18 and Inception-v3 using the same classification task; the corresponding results are included in Appendix G (see Fig. 8 and Fig. 9). Moreover, in the rebuttal period, we also provide further experimental results in this response for completeness (see in new uploaded supplementary materials).

---

### Official Review · Reviewer_UY6e · 2025-11-03

**Soundness:** 3
**Presentation:** 2
**Contribution:** 1
**Rating:** 2
**Confidence:** 5

**Summary:**

The paper establishes the result on direction of GD convergence to the minima (with the $2/(\lambda_1 + \lambda_n)$ as the threshold between existence of direction and direction alternating at every step. A perturbative explanation for sharpness oscillations is given. Experiments conducted to show a presence of oscillations in the NN dynamics under GD/SGD/Adam.

**Strengths:**

The proof is correct and carefully written, adding the control for higher-order terms; the experiments are clear in what they are measuring and well-explained.

**Weaknesses:**

Theory:
    - unfortunately, there is no real novelty in the proven theoretical result (Theorem 1), as it is known in the classical iterative methods literature, see e.g. [1], Chapter 4. That is, your result is known for quadratics, the nonlinear extension is a technicality.
    - Considering that this is the main part of the paper (see the details about empirical part below), this lack of novelty constitutes enough of a ground for rejection.
Connection to EoS:
    - The above phenomenon is not really related to EoS. EoS happens also (and specifically!) when dynamics are away from minima - and that’s almost the point of it, as the training continues as the network is in EoS (see original Cohen’s EoS, 2021). Your result is asymptotic near a strict minima, which makes them only connected because both are about oscillations, and that’s it
    - there they are always zero (and negative) eigenvalues present in the spectrum of the Hessian of NN, see [2] or [3]. This makes the bound $2/(\lambda_1 + \lambda_n) = 2/\lambda_n$, therefore eliminating the regime you term “large learning rate regime”
    - Considering that you are discussing the oscillations of top eigenvalue during the EoS oscillations, you should contrast your work with Damian et al. “Self-Stabilization…” (2022), as this is precisely what this work is doing - showing how and why the sharpness oscillates and produced the self-stabilization
Empirics:
    - the empirics are basically the experiments of “A Walk with SGD” of Xing et al. (2018), albeit with Adam added. They are measuring the exact same quantity, showing the oscillations. Therefore, there is no novelty in the experiments either. In particular, the GD experiments show the EoS phenomena (as later described by Cohen et al al. in EoS paper)
    - Moreover, the experiments are unrelated to the theorem you are proving, see the above point about EoS being away from minima — that is, the oscillations start when we are away from minima, and there is no inherent reason why the oscillations in the experiments are the same oscillations that are talked about in the theorem. Therefore, it is unclear how the experiments support your finding.

**Questions:**

- What is the difference between your experiments and those of Xing et al.?
- How does your sharpness oscillations results compare to those of Damian et al. “Self-Stabilization…”

---

> ### Author Response · Authors · 2025-11-26
> **Rebuttal for Comments from Reviewer UY6e**
>
> >Unfortunately, there is no real novelty in the proven theoretical result (Theorem 1), as it is known in the classical iterative methods literature, see e.g. [1], Chapter 4. That is, your result is known for quadratics, the nonlinear extension is a technicality.
>
> We thank for reviewer's comments.
>
> First, while classical iterative methods indeed characterize the dynamics of GD on quadratic losses, it is well established that the EoS phenomenon does not occur for quadratic objectives. The novelty of our work lies precisely in establishing a rigorous theoretical result that controls higher-order terms beyond the quadratic approximation, thereby allowing a mathematically justified analysis of the local dynamics of GD near minima.
>
> Moreover, the nonlinear extension is not merely a technicality. It is essential for understanding the mechanisms underlying the transition from EoS-like oscillatory behavior to asymptotic convergence. To the best of our knowledge, no existing work provides such a formal, rigorous result for this regime.
>
> Finally, we note that the reviewer does not specify which precise reference [1].
>
> > The above phenomenon is not really related to EoS. EoS happens when dynamics are away from minima… your result is asymptotic near a strict minimum… only connected because both are about oscillations.
>
> We appreciate the reviewer's comments and would like to clarify that the EoS phenomenon encompasses a variety of dynamical behaviors, including transitions near and away from minima. Both early-phase and late-stage dynamics are focused and concerned in recent works, while our work investigates the latter one. Hence, our theoretical framework contributes to understanding one possible mechanism connecting EoS-like oscillations to eventual convergence.
>
> >There are always zero (and negative) eigenvalues in NN Hessians… therefore eliminating the ``large learning rate regime'' you describe.
>
> We thank the reviewer for raising this point. While the Hessian of a neural network may generally contain zero or negative eigenvalues, the situations involving negative curvature are not relevant to the setting we study, since such points cannot be minimum. For the case of zero eigenvalues, degenerate critical points such as saddle points may arise. These require additional considerations are too sophisticated that are beyond the scope of the present work. To keep the analysis focused and technically tractable, we restrict attention to the case near locally strongly convex minimums.

---

> ### Author Response · Authors · 2025-11-26
> **Rebuttal for Comments from Reviewer UY6e (Continue)**
>
> >You should contrast your work with Damian et al., ``Self-Stabilization…'' (2022).
>
> >How do your sharpness oscillation results compare to those of Damian et al.\ (2023)?
>
> We appreciate the reviewer's suggestion and have clarified the relationship in the response below.
> In the following we summarize the elements of Damian et al. (2022) that are most relevant for comparison, and then highlight the differences and complementarity between their framework and ours.
>
> **Summary of Damian et al. (2022).**
>
> The paper is aimed specifically at explaining the behaviour in the EoS regime, where sharpness approaches and oscillates around the threshold $2/\eta$ (see their Figure 1). Their theoretical development concentrates on the mechanism of progressive sharpening and sharpness oscillations within this regime.
>
> At first, they propose central component of their analysis, which is Lemma 2, (under simplicity of the top Hessian eigenvalue) establishes
> $$\nabla S(\theta) = \nabla^3 L(\theta)\bigl(u(\theta),u(\theta)\bigr),$$
> where $S(\theta)$ denotes the sharpness at point $\theta$ (the maximum eigenvalue of $\nabla^2 L(\theta)$), $L(\theta)$ is the loss function and $u(\theta)$ is the eigenvector w.r.t. $S(\theta)$.
> This identity connects the evolution of sharpness to third-order derivatives and underlies their explanation of instability and stabilization along the top-curvature direction.
>
> Then, to facilitate analysis, the authors introduce the projected gradient descent (PGD) trajectory
> $$\theta_{t+1}^\dagger = \operatorname{proj}_{\mathcal{M}}\bigl(\theta_t^\dagger - \eta \nabla L(\theta_t^\dagger)\bigr), \qquad \mathcal{M}=\{\theta : S(\theta)\le 2/\eta, \nabla L(\theta)\cdot u(\theta)=0\},$$
> and compare GD iterates to this reference path.
>
> Their time index is shifted so that analysis begins at the first point $\theta_0$ satisfying
> $$S\bigl(\operatorname{proj}_{\mathcal{M}}(\theta_0)\bigr)=2/\eta,$$
> i.e. the dynamics have already reached the EoS boundary.
>
> Their Assumption 1 assumes that
> $$\alpha(\theta) = -\nabla L(\theta)\cdot \nabla S(\theta) > 0$$
> along the PGD path. This corresponds to persistent progressive sharpening, although in practice the sign of this quantity may vary throughout training.
> Actually, for the case of gradient flow, i.e.,
> $$\dot{\theta}(t)=-\nabla L(\theta(t)).$$
> Then one has,
> $$(S(\theta(t)))^\prime = \left< \nabla S(\theta(t)), \dot{\theta}(t) \right>=-\left< \nabla S(\theta(t)), \nabla L(\theta(t)) \right> = \alpha(\theta(t)),$$
> this indicates that the sharpness is monotonic increasing if one assumes that $\alpha(\theta)>0$.
>
> Then, in Section 4, the authors introduce a two-dimensional reduced system in variables $(x_t, y_t)$ and describe a four-stage cycle that illustrates the sharpness oscillation in the EoS regime.
>
> Moreover, in Section 5, while introducing structural Assumptions 4 and 5 on third derivatives, spectral separation, and subspace regularity. These support their Theorem 1, which generalizes the reduced-cycle explanation. The conclusions rely on these assumptions and are supported by experiments.
>
> Overall, Damian et al. develop a reduced-model explanation tailored to the EoS regime, relying on explicit structural assumptions introduced in Sections 4-5.
>
> **How our work differs and how the results relate.**
>
> **1. Different regime and objective.**
> Our work focuses on the regime in which GD converges. The convergent behaviour can be guaranteed when $\eta < 2/\lambda_{\max}$ (see Chen et al., "On the Unstable Convergence Regime of Gradient Descent" AAAI'24). We provide a rigorous analysis of the existence of a convergence direction in this stage. While it is difficult to determine precisely when GD exits EoS in neural network training, our results can be considered as a reference that GD may begin to exhibit convergent behaviour.
>
> **2. Different sharpness behaviour near the limit point.**
> As GD approaches convergence, our analysis shows that $\lambda_{\max}(x_k)\to \lambda_{\max}(x^*)$.
>
> For large learning rates, the sharpness may oscillate around $\lambda_{\max}(x^*)$, which may not equal to $2/\eta$.
>
> This behaviour is fundamentally different from that described by Damian et al., where sharpness oscillates around $2/\eta$ during the EoS stage. Thus our results describe behaviour after the EoS phase, and do not conflict with their characterization of the EoS regime.
>
> Finally, as a conclusion, Damian et al. present a reduced-model description for EoS regime under several structural assumptions. Our work analyzes the subsequent phase, which GD approaches to convergence and rigorously characterizes GD dynamics under assumptions suited to this stage. The two analyses address different parts of the training trajectory and are therefore complementary rather than contradictory.

---

> ### Author Response · Authors · 2025-11-26
> **Rebuttal for Comments from Reviewer UY6e (Part 3)**
>
> >The experiments are similar to ``A Walk with SGD'' (Xing et al., 2018)… no novelty… and unrelated to the theorem.
>
> >What is the difference between your experiments and those of Xing et al.?
>
> We thank the reviewer for this observation.
> Our experiments are not intended as novel empirical discoveries, but rather as illustrative demonstrations of the theoretical implications. Specifically, we visualize how the limit direction and local oscillation behavior predicted by our theorem manifest under different optimizers. While the setup indeed resembles that of Xing et al., our focus and interpretation differ: we examine the emergence of the limit direction of GD that may apply to more gradient methods, and our experiments show three typical and widely used algorithms.

---

### Meta-Review · Area_Chair_ELJ6 · 2026-01-07

**Summary:**

This theoretical work gives a novel insight into the asymptotic direction of GD convergence near an isolated local minimum with a convexity. It reveals a threshold of the learning rate separating two regimes: one in which the iterates of gradient descent converge in a single direction, and the other in which they show an alternating convergence direction. The authors discuss the relation of this finding to EoS phenomena.

The reviewers are clearly divided into two sides. As the reviewers on the acceptance side evaluated, I think that the finding of this switching of convergence dynamics depending on $\eta$ itself is highly interesting, from a purely theoretical curiosity perspective.
As the reject-side reviewers pointed out, however, the problem is how this finding is essentially related to EoS or to other optimization (or ML) problems. That is, the current finding has some properties that differ from other theoretical or empirical observations of EoS.
To this, in the rebuttal, the authors replied that the understanding of EoS is discussed only as a suggestion.

Their finding itself is very solid and interesting as a fundamental property of optimization, and this will enrich and move forward our understanding of gradient descent, including the fact that it has triggered a discussion on whether or not this is related to the EoS.
Thus, following the accept-side reviewers, I evaluated this work as acceptable.

I strongly recommend the authors to revise some misleading sentences in the abstract and the introduction, which make the readers think that this work contributes to the understanding of the EoS. The potential connection to the EoS should be explained not as a contribution (like item of Line 115),  but only as a suggestion, whose details are explained in the final part of the manuscript.

**Reviewer Concerns:**

The following concerns show that more exhaustive experiments or theoretical justification will be necessary to relate the current finding with EoS:

**Reviewer UY6e**
- EoS happens also (and specifically!) when dynamics are away from minima
-  there they are always zero (and negative) eigenvalues present in the spectrum of the Hessian of NN.
-  ... the oscillations start when we are away from minima, and there is no inherent reason why the oscillations in the experiments are the same oscillations that are talked about in the theorem

**Reviewer YPEn**
- (ii) In this setting, even if the sharpness exhibit oscillations, the loss still decreases monotonically, as it seems to be observed on figure 2.c. This is not consistent with the typical behavior of the EoS regime
- (iii) According to [1], ... growth is more likely to be sub linear. Such a setting is not covered by the implicit "local strong convexity" assumption of the authors.

However, if the authors fix the manuscript so that the connection to EoS is not presented as a contribution but as a suggestion of this paper, these point-outs are not major concerns.

**Reviewer Scores:**

Reviewers UY6e and YPEn could increase their scores if they understand that this study should be positioned not as a contribution to the EoS, but as a study on a fundamental property of gradient descent.

---

### Decision · Program_Chairs · 2026-01-26

Accept (Poster)